# Analysis of the economic burden of diagnosis and treatment on patients with tuberculosis in Bao'an district of Shenzhen City, China

**Yixiang Huang**[1☯], **Jianying Huang**[2☯], **Xiaoting Su**[1], **Liang Chen**[3], **Jianwei Guo**[1], **Weiqing Chen**[4]*, **Lingling Zhang**[5]

**1** Department of Health Policy and Management, School of Public Health, Sun Yat-sen University, Guangzhou, China, **2** Guangdong Women and Children Hospital, Guangzhou, China, **3** Centre for Tuberculosis Control of Guangdong Province, Guangzhou, China, **4** Department of Medical Statistics and Epidemiology, School of Public Health, Sun Yat-sen University, Guangzhou, China, **5** Department of Nursing, College of Nursing and Health Sciences, University of Massachusetts, Boston, United States of America

☯ These authors contributed equally to this work.
* chenwq@mail.sysu.edu.cn

## Abstract

### Background

Illness-related costs experienced by tuberculosis patients produce a severe economic impact on households, especially poor families. Few studies have investigated the full costs, including direct and indirect costs, at the patient and household levels in south-east China.

### Methods

A case follow-up study was conducted in the Bao'an district of Shenzhen City, China. Eligible new and previously treated individuals with pulmonary tuberculosis (TB) during January 1st 2013 to June 30th 2013 were enrolled. Medical and non-medical costs as well as income loss were calculated in diagnosis and treatment periods, respectively. Factors associated with costs due to TB diagnosis, treatment and TB care (diagnosis + treatment) were explored respectively with a linear regression model.

### Results

Of the total 514 TB patients enrolled, 95% were from the migrant population, and 65% were males, with a mean age of 32.25 (±10.11). The median costs due to TB diagnosis and TB treatment were 79 United States dollar (USD), 748USD (6.2897 China Yuan (CNY) = 1USD, 2013) per patient, respectively. The median costs due to TB care (diagnosis and treatment) per patient was 1218USD, corresponding to 26% of patients' annual income pre-illness. Those who visited more times to health facilities, hospitalized, received higher education, or occupied in national civil servant/services/retired staff might expense more before diagnosis. Costs due to TB treatment was significantly higher among migrant patients, sputum smear positive patients, and widowed/divorced population. Factors associated with less total costs were native patients, fewer times of visiting to health-care facilities and those with no hospitalization history due to TB.

**Data Availability Statement:** All relevant data are within the manuscript and its Supporting Information files.

**Funding:** This project was funded by the National Major Science and Technology Programs in the "Twelfth Five-Year" Plan period (No. 2012ZX10004903). The funders had no role in study design, data collection and analysis, decision to publish, or preparation of the manuscript.

**Competing interests:** The authors have declared that no competing interests exist.

## Conclusions

Although a free TB control policy is in force, patients with TB are still facing a heavy economic burden. More available interventions to reduce the financial burden on tuberculosis patients are urgently needed.

## 1. Background

Tuberculosis (TB) is one of the top 10 causes of death worldwide as well as being the leading cause from a single infectious agent (more than HIV/AIDS). According to 2018 global estimates, there were estimated 10 million new cases of TB and 1.2 million TB deaths in 2018 [1]. The high prevalence of TB is always a critical public health problem in China: China has the second highest number of cases of TB in the world right after India, with 866,000 identified cases in 2018 [1]. Since 1990, the World Health Organization(WHO)-recommended directly observed treatment + short course chemotherapy (DOTS) strategy has been implemented by the Chinese government under the National Tuberculosis Control Program to control this epidemic [2].

Many investigators have pointed out that both the TB patients and their households bear a heavy economic burden under the free DOTS control strategy [3, 4]. TB is a well-known poverty-related disease, usually related to the disadvantages of socio-economic status. In particular, among those who come from rural and poor districts, the scarcity of health resources and poor economic condition place them in a more serious situation [5]. Delayed and repeated visits to clinics before diagnosis, the over-prescribing of drugs, and prolonged treatment are common [6]. Patients bear too much economic burden which precludes them from further adhering to treatment and results in treatment failure [4].

There have been many studies on the cost-effectiveness of different TB control strategies, focused mainly on the effectiveness of medical insurance for coping with the costs of illness as well as patient compliance with drug regimens [4, 7]. However, far fewer studies have measured the costs to patients and households or the full costs, including direct and indirect costs, from the onset of symptoms to the end of treatment in south-east China. Therefore, the present study was performed to evaluate the out-of-pocket direct costs as well as productivity lost related to illness. Potential factors associated with these costs were also explored.

## 2. Materials and methods

### 2.1. Study setting

Shenzhen City is one of the most prosperous regions in China, with a Gross Domestic Product (GDP) of nearly 1450 billion CNY in 2013. Its estimated population is over 10.5 million, with the migrant population constituting72.7% while this rate is 87.0% in Bao'an district of the total regional population according to the Shenzhen statistical yearbook 2013. The mean household income in the region was 6,656.79 USD in 2012 according to the statistical yearbook 2013. The rapid economic growth in the city attracts numerous young migrant workers, mostly from remote poor areas exposed to substandard conditions such as lower societal status, poor standard of living, and less accessibility to health services, thus creating favorable conditions for infection with TB. Although analysis of recent data has shown a decreasing incidence of TB in the Shenzhen region, its prevalence in the Bao'an district, where a heavy concentration of migrant workers contributes to a high TB case load, is still high.

The general hospitals were the main place to find TB cases and the Center for Prevention and Cure of chronic diseases (CPC) were the local designated TB facilities in the national TB program networks, providing treatment and managing the TB cases. The health-care providers were obligate to refer the suspected cases to CPC and the basic information of suspected cases were sent to CPC. If the cases did not consult to CPC in time, the workers of CPC would contact the suspected case. In 2010, the prevalence rate of smear positive of native population was 19.44 per 100,000, and that of the migrant population was 59.96 per 100,000, and the cure rates for native and migrant patients were 94.79% and 82.91%, respectively [8].

The DOTS program is implemented by public health facilities for non-inpatients TB cases [9], where patients have free access to diagnostic and treatment services. The CPC is the institution authorized to provide TB diagnosis, treatment, and monitoring, with a radiologic imaging studies (X-ray) and sputum smear tests while diagnosed for free, and 5 sputum smear tests and anti-TB drugs for 6–8 months during treatment for free. Patients must regularly return to CPC every month for health checks after starting to take anti-TB drugs until the treatment is over. Generally, the numbers of times of subsequent consultations are 5 and 7 to newly-patents and previously treated patients respectively. Except free TB drugs provided by CPC, the subsidiary drugs like hepatinica, gastric medicine, and immunopotentiator etc. prescribed every month and auxiliary examinations including X-ray and hepatic, renal function tests. are not free.

## 2.2. Study design and population

A case follow-up study was conducted in the Bao'an district with an ongoing NTP-DOTS project. Newly diagnosed and previously treated TB patients (not known to be drug resistant) aged 15–59 years old, who registered in the CPC during 1 January 2013 to 30 June 2013, were included and were followed up until the completion of a 6- to 8-month treatment course. Those with severe co-morbidities such as AIDS, diabetes, lung cancer and those who refused to accept investigation were excluded. Patients aged above 59 were excluded considering the elderly may have more respiratory system diseases affecting the result of our research and those with MDR-TB were excluded too. Transferred-in patients were also excluded from the study, since information on previous costs was not available.

## 2.3. Data collection

All eligible cases would undergo five (new patients) or seven (previously treated patients) interviews at the Department of Tuberculosis Prevention and Control in Bao'an CPC. A structured questionnaire was used, covering general demographic and socio-economic characteristics, disease history, care-seeking process, treatment behaviors, and costs due to TB diagnosis and treatment. Self-reported economic status information was collected based on different localities with various levels of development. The first interview took place at the first month of intensive phase of treatment, when patients' retrospective care-seeking history and related expenditures, including costs due to TB diagnosis starting from the time point when the patients experienced the first symptoms, and the intensive phase related costs, were collected. The other four or six interviews were conducted monthly when patients came for regular re-examinations. The patients' TB care experiences and transportation fee, supplementary food cost, and foregone income of patients and companions were collected in follow-ups.

A graduate student with two locally recruited research assistants, health workers who had participated in a standardized training session, conducted those interviews. A logic check of all data collected was undertaken to determine if there were any contradictions or missing information. Those contradictions or missing information would be asked again in a subsequent

interview. We also evaluated the quality of collected data by verifying the routine health information system.

## 2.4. Measurements and definitions

We analyzed the economic burden from TB patients' perspective. The primary outcome variables were the median expense incurred during illness. Information on costs was ascertained for different periods, summarized as costs due to TB diagnosis, TB treatment and TB care (diagnosis and treatment).

The operational definition of various costs and indicators were summarized in Table 1. Direct costs included all costs of patients and companions attributable to their illness [10]. Indirect costs referred to the income lost by patients and their companions associated with time lost off work. The income lost before diagnosis was calculated based on each patient's monthly wage prior to the onset of TB. The income lost by companions before diagnosis was not included in the analysis, since only a few such losses occurred. The time spent on daily drug intake was also not calculated, because 441(85.80%) of patients took drugs on their way to work or the community health-care stations close to their homes by walk or bike. The costs due to TB diagnosis of those who were diagnosed at CPC immediately without consulting to other facilities, were not calculated.

The economic burden was then analyzed as a proportion of patients' total annual income. The free services provided during diagnosis and treatments were not included in the medical costs. Information about costs after diagnosis was extracted from the information system of the study facility. Information on costs covered by insurance could not be obtained, since the insurance had not been settled at the end of the interview.

**Table 1. Operational definition of study participants and TB treatment cost in the study of Bao'an district, Shenzhen City, China, 2013.**

| Terminology | Definition |
|---|---|
| Costs due to TB diagnosis | Costs between symptom onset and diagnosed as TB |
| Costs due to TB Treatment | Costs from treatment initiation up to treatment completion |
| Direct medical costs | Expenses of medical examinations and medicines linked to TB diagnosis and treatment |
| Direct non-medical costs | Costs for transport, food expenditures, nutrition supplements due to TB |
| Direct cost | Direct medical costs (clinical and hospitalization expenses) + direct non-medical costs (transportation to health facilities and supplementary food) |
| Indirect costs | Patients and companions' lost income due to TB-related time off work during the TB episode. Assuming 30 working days per month, we evaluated the value in terms of money for each day, and indirect costs before diagnosis was calculated as the value per day multiplied by the length of time off work due to illness. Patients were asked about the actual lost income that they and their companions had experienced due to absence from their usual income-generating activities at every interview. These monthly income reductions were then summed as indirect costs during TB treatment. |
| Costs due to TB care (diagnosis and treatment) | Direct costs + indirect costs |
| The degree of symptoms at diagnosis | The patients self-reported with different symptoms at diagnosis were divided into four categories. 1. None: no symptoms; 2. mild: with symptoms included cough, expectoration and dyspnea, night sweat, and debilitation; 3. Moderate: symptoms with fever and chest distress; 4. severe: symptoms like hemoptysis. |
| Interruption of treatment | Treatment interruption is defined as any interruption of treatment for at least one day but for < 8 consecutive weeks |

## 2.5. Data analysis

The data were double-entered with EpiData 3.1 software (EpiData Association, Odense, Denmark) for each patient, and cost summaries and analyses were performed by SPSS 20.0 software. Continuous variables such as direct costs, indirect costs were summarized as medians (IQR), while categorical variables were summarized as proportions. A linear regression (forward stepwise) was used to build a predictive model for factors related to costs due to TB diagnosis, treatment and care respectively. The costs were log transformed as they were not normally distributed. Association was summarized using Beta coefficient (95% CI), *P*-value less than 0.05 were significant. Considering the multicollinearity in predictor variables, we calculated variance inflation factor (VIF) at the same time [11]. If VIF was less than 10, then the predictor variables were included in the linear model.

Considering that applying and interpreting the results of hypothesis testing in a log-transformed data on actual data (non-log transformed) might lead to bias [12], we also conducted a confounder-adjusted association between costs and predictor variables using generalized linear model (logistic regression). The costs were divided as binomial variables based on median value. Association was summarized using coefficient (95% CI), P-value less than 0.05 was significant.

## 2.6. Ethics approval

This study was approved by the research and ethics committee of the School of Public Health, Sun Yat-Sen University, China. The ethical approval number was 2012-SPH021, date 9/28/2012. Also, permission was obtained from the study site. A signed informed consent was obtained from all patients before being interviewed.

## 3. Results

### 3.1. Socio-demographic characteristics and care-seeking behaviors

The present study surveyed 533 eligible TB patients with no co-morbidity, 19 (4%) of whom were lost to follow-up, one of whom died from another disease, and 18 of whom moved to other provinces. The average age of them was 29.6±10.2 years, with 14 males and 5 females. One patient suffered from hemoptysis, which was defined as a severe symptom in our study. The others were suffered from mild or moderate symptoms like cough, chest distress and fever etc. The socio-demographic and care-seeking behaviors characteristics of 514 patients were summarized in Table 2. The mean monthly income of patients before illness was 448USD. 34%(176) patients were identified when they accepted health check-ups required by their work units, and were then transferred to the CPC.

### 3.2. Costs due to TB diagnosis, TB treatment and TB care

The costs and component of different of TB episode were showing in Tables 3 and 4. 338 patients incurred costs due to consultation before diagnosed as TB. While 22 patients were admitted to the hospital. The direct costs accounted for 96.08% of total costs due to TB diagnosis, while accounted for 57.9% during TB treatment (Table 4).

The main component of medical costs during TB treatment was examinations and non-TB drugs (liver/kidney protection drugs and immunopotentiators), accounting for 55% and 40% of medical costs. The direct costs of intensive phase (the first two month) and consolidation phase (the next 4 or 6 month) were 210.58(164.87,279.66)USD and 516.48(391.91,703.39) USD respectively. The direct costs of consolidation phase accounted for 70.7% of total costs due to TB treatment.

**Table 2. Baseline characteristics of TB patients enrolled in the study of Bao'an district, Shenzhen, China, 2013 (N = 514).**

| Variable | Number | % |
|---|---|---|
| Age (years) | | |
| < 20 | 20 | 3.89 |
| 20~30 | 238 | 46.3 |
| 30~40 | 143 | 27.82 |
| 40~50 | 70 | 13.62 |
| >50 | 43 | 8.37 |
| Gender | | |
| Male | 336 | 65.37 |
| Female | 178 | 34.63 |
| Marital status | | |
| Single | 180 | 35.02 |
| Married | 308 | 59.92 |
| Other (divorced, cohabiting) | 26 | 5.06 |
| Education | | |
| Illiterate or primary school | 63 | 12.26 |
| Junior high school | 220 | 42.80 |
| Senior high school | 169 | 32.88 |
| Junior college and above | 62 | 12.06 |
| Occupation | | |
| Factory worker | 316 | 61.48 |
| Individual business | 46 | 8.95 |
| Other (include company stuff/service personnel) | 99 | 19.26 |
| Unemployed | 53 | 10.31 |
| Residence | | |
| Native | 26 | 5.06 |
| Migrant | 488 | 94.94 |
| Self-reported economic status* | | |
| High | 30 | 5.84 |
| Medium | 323 | 62.84 |
| Low | 161 | 31.32 |
| Annual household income (USD) | | |
| <795 | 13 | 2.53 |
| 795–2385 | 44 | 8.56 |
| 2385–4770 | 114 | 22.18 |
| 4770–9539 | 133 | 25.88 |
| 9539–15,899 | 113 | 21.98 |
| >15,899 | 97 | 18.87 |
| Insurance | | |
| Labor insurance | 186 | 36.19 |
| Self-paying | 305 | 59.34 |
| Unclear | 23 | 4.47 |
| First visited a public facility | | |
| Yes | 48 | 9.34 |
| No | 466 | 90.66 |
| Type of TB | | |
| Newly diagnosed patients | 484 | 94.16 |

(*Continued*)

**Table 2.** (Continued)

| Variable | Number | % |
|---|---|---|
| Previously treated patients | 30 | 5.84 |
| The way of discovering TB presence | | |
| Health check-up | 176 | 34.24 |
| Visiting doctors because of developing symptoms | 338 | 65.76 |
| Number of times visiting health-care facilities | | |
| < = 2 | 214 | 63.31 |
| 3~6 | 101 | 29.88 |
| > = 7 | 23 | 6.8 |
| Delay between symptom onset and first consultation | | |
| < = 7 | 152 | 44.97 |
| 7~14 | 38 | 11.24 |
| 14~21 | 42 | 12.43 |
| > = 21 | 106 | 31.36 |

* Self-reported economic status was based on the local condition of patients' domiciles.

The median costs due to TB care was 1218USD (Table 3), which corresponds to 26%(1218/4685) of annual individual incomes before TB illness (4770USD). The direct costs accounted for 61% of the total costs, and the ratio of direct to indirect costs was 1.58%. About 88.2% of costs were encountered after the patients were diagnosed and accepting free DOTS treatment.

### 3.3. The association of costs due to TB diagnosis among TB patients

Patients who were in hospital due to TB generated more costs ($\beta$ = -1.13, P<0.0001)(Table 5). It seemed that the less number of times visiting to health-care facilities (< = 2 vs > = 7,$\beta$ = -0.95, P<0.0001; 2~6 vs > = 7, $\beta$ = -0.46, P<0.0001), the less costs incurred.

We found those who received higher education($\beta$ = 0.24,P = 0.007) or occupied in national civil servant/services /retired staff ($\beta$ = 0.12,P = 0.046)might expense more before diagnosis compared with the primary/illiterate and factory workers. In the corresponding generalized linear regression, number of times visiting health-care facilities ($\beta$ = 4.17,P<0.001)and education were found to be related with costs due to diagnosis statically significant. Gender($\beta$ = 0.78, P = 0.021) was found to be an independent factor related costs due to TB diagnosis in the logistic regression (S1 Table).

**Table 3. Costs due to TB diagnosis, treatment and care (diagnosis and treatment) among TB patients during different episodes of treatment in the study of Bao'an district, Shenzhen City, China, 2013(USD).**

| Costs due to TB diagnosis and treatment | TB diagnosis (N = 514) | TB treatment (N = 514) | TB care(N = 514) |
|---|---|---|---|
| | Median (IQR) | Median (IQR) | Median (IQR) |
| Direct medical costs | 23.6(0–99.4) | 5421.5(464.7–617.8) | 600.4(510.1–734.9) |
| Direct non-medical costs(travel) | 0 (0–3.2) | 149.9(29.3–453.4) | 156.5(33.7–411.7) |
| Direct costs(all) | 23.8 (0–111.3) | 748.5 (567.4–986.8) | 833 .4(608.9–1132.3) |
| Indirect costs (wage/income lost) | 0 (0.0–0.0) | 151.8(50.4–844.6) | 157.4 (56.8–854.5) |
| Total costs | 23.8 (0–113.0) | 1074.5 (744.5–1819.6) | 1218.0 (826.1–1963.4) |

IQR: Interquartile range.

[a]Average China Yuan (CNY) to USD conversion rate in 2013 (1USD = 6.2897 CNY).

**Table 4. Contribution of each component of costs due to TB diagnosis/treatment/care as a proportion of total costs among TB patients during different episodes of treatment in the study of Bao'an district, Shenzhen City, China, 2013.**

| Costs due to TB diagnosis and treatment as a proportion of total costs | TB diagnosis(N = 514) | TB treatment(N = 514) | TB care (N = 514) |
|---|---|---|---|
| | % | % | % |
| Total costs | 100 | 100 | 100 |
| Direct medical costs | 93.12 | 38.43 | 43.44 |
| Direct non-medical costs(travel) | 2.96 | 19.47 | 17.96 |
| Direct costs(all) | 96.08 | 57.90 | 61.40 |
| Indirect costs (wage/income lost) | 3.92 | 42.10 | 38.60 |

### 3.4 The association of costs due to TB treatment among TB patients

Final model for independent predictors for costs due to TB treatment was shown in Table 6. Migrant population might have more costs during the TB treatment compared with native population (β = 0.15,P = 0.006). Those who widowed/divorced(β = 0.17,P = 0.049) incurred more costs after diagnosis as TB. Patients perceived that the TB imposed economic burden to household might have more expenses on the treatment r(β = -0.07,P = 0.043)).The status of sputum smear test at the time of diagnosis was positive generated more costs compared to the negative (β = 0.08,P = 0.021).

In the corresponding generalized linear regression, only sputum smear status adopted in the model. (P<0.0001). (S2 Table).

### 3.5 The association of costs due to TB care (diagnosis and treatment) among TB patients

Final model for independent predictors for costs due to TB care(diagnosis + treatment) was shown in Table 7. Patients with no history of hospitalization(β = -0.26,P<0.001) and fewer

**Table 5. Confounder adjusted association between costs due to TB diagnosis and various predictor variables using linear regression models in the study of Bao'an district, Shenzhen City, China, 2013 (n = 514).**

| Predictors in the model | Beta coefficient | (95% CI) | P value |
|---|---|---|---|
| Number of times visiting health-care facilities | | | |
| < = 2 | -0.95 | -1.18,-0.73 | < 0.001 |
| 2~6 | -0.46 | -0.65,-0.27 | < 0.001 |
| > = 7 | Ref | Ref | Ref |
| Whether in hospital due to TB diagnosis | | | |
| Yes | Ref | Ref | Ref |
| No | -1.13 | -1.14,-0.95 | < 0.001 |
| Education | | | |
| Primary/illiterate | Ref | Ref | Ref |
| Junior high school | 0.10 | -0.03,0.24 | 0.139 |
| Senior high school | 0.11 | -0.04,0.25 | 0.146 |
| College or above | 0.24 | 0.07,0.41 | 0.007 |
| Occupation | | | |
| Workers | Ref | Ref | Ref |
| Individual business | 0.026 | -0.12,0.17 | 0.731 |
| Others national civil servant/services /retired staff etc.) | 0.12 | 0.002,0.24 | 0.046 |
| Unemployed | -0.07 | -0.22,0.8 | 0.341 |

Linear regression was done after conversion to log scale.

Only significant variables were presented.

Ref-reference.

**Table 6. Confounder adjusted association between costs due to TB treatment and various predictor variables using linear regression models in the study of Bao'an district, Shenzhen City, China, 2013 (n = 514).**

| Predictors in the model | Beta coefficient | (95% CI) | P value |
|---|---|---|---|
| Sputum smear status | | | |
| Negative | Ref | Ref | Ref |
| Positive | 0.08 | (0.036,0.128) | < 0.001 |
| Household registration | | | |
| Native patients | Ref | Ref | Ref |
| Migrant patients | 0.15 | (0.043–0.254) | 0.006 |
| Reported-household economic burden | | | |
| Heavy | Ref | Ref | Ref |
| Moderate | -0.07 | (-0.13,-0.02) | 0.043 |
| No burden | -0.01 | (-0.07,-0.05 | 0.73 |
| Marital status | | | |
| Unmarried | Ref | Ref | Ref |
| Married | -0.005 | (-0.06,0.05) | 0.860 |
| Widowed/divorced | 0.17 | (0.01,0.34) | 0.049 |
| Others | -0.02 | (-0.15,0.12) | 0.820 |

Linear regression was done after conversion to log scale.

Only significant variables were presented.

Ref-reference.

visits to health facilities ($\beta$ = -0.15,P = 0.011) were independent predictors for less costs. The native people ($\beta$ = 0.13,P = 0.038) incurred less costs of TB care. Sputum smear status and number of times visiting health-care facilities were also included in the model in the corresponding logistic regression (S3 Table).

## 4. Discussion

### 4.1 Summary of key findings

This study evaluated the economic burden borne by TB patients in Shenzhen City during their illness and explored the influence factors related to the costs. There were some key findings

**Table 7. Confounder adjusted association between costs due to TB care (diagnosis and treatment) and various predictor variables using linear regression models in the study of Bao'an district, Shenzhen City, China, 2013 (n = 514).**

| Predictors in the model | Beta coefficient | 95% CI | P value |
|---|---|---|---|
| Whether in hospital due to TB diagnosis | | | |
| Yes | -0.26 | -0.38,-0.15 | < 0.001 |
| No | Ref | Ref | Ref |
| Household registration | | | |
| Native patients | Ref | Ref | Ref |
| Migrant patients | 0.13 | 0.01,0.26 | 0.038 |
| Number of times visiting health-care facilities | | | |
| < = 2 | -0.15 | -0.26,-0.03 | 0.011 |
| 2~6 | -0.1 | -0.22,0.02 | 0.094 |
| > = 7 | Ref | Ref | Ref |

Linear regression was done after conversion to log scale.

Only significant variables were presented.

Ref-reference.

from this study. The tuberculosis patients still faced a high economic burden during TB diagnosis and treatment. Direct costs accounted a larger proportion of total costs. More costs occurred during TB treatment when accepting free DOTS treatment. Those who visited more times to health facilities, hospitalized, received higher education, or occupied in national civil servant/services/retired staff might expense more before diagnosis. Costs due to TB treatment was significantly higher among migrant patients, sputum smear positive patients. Native patients, those with fewer times of visiting to health-care facilities and with on hospitalization history due to TB costs less.

## 4.2 Discussion of key findings

Under the DOTS program, the CPC provides free treatments amounting to 102.07USD. The median cost due to TB care was 1218USD, corresponding to 26% of annual personal income, constituting a considerable part of their disposable income after payment of fixed costs such as rent, electricity, and water etc. The total costs found in this study were far higher than those in Africa, India, and rural China [13–15]. These differences could partly be explained by the differences in purchasing power among these countries/districts, the omission of some cost elements in these studies or reduced recall bias in our study. Over a quarter of patients reported that the economic burden of cure was heavy. Some authors have concluded that total costs due to TB care account for a higher percentage of their annual income [13, 16–18].

Direct costs were still substantial, of which non-TB drugs and auxiliary examinations were the most significant cost items [19–21]. Additional drugs like protecting the liver/kidney or symptomatic treatment, and examinations must be paid for. It appeared that these costs led to the high costs during the treatment period, indicating that there might be over-prescription and over-service, as reported in other studies [22, 23]. About 37% (190/514) of the patients were unemployed or suspended from work after contracting TB and had no income, while the others remained employed and experienced income loss due to monthly doctor visits. Thus, departure from daily money-earning activities had a negative economic impact on patients with TB, resulting in impoverishment and worsening of their living situations [19]. However, the indirect costs in these cases might be underestimated, since the value of time loss due to daily drug intake was not calculated, and some patients reported that they had no clear concept of their reduced income due to absences from the workplace.

Costs due to TB diagnosis were much lower compared with the costs during TB treatment. This is consistent with reports from other studies [24, 25]. A review of the financial burden of TB indicated that half of total costs were incurred before treatment [26]. The possible reason may be that many patients in this study were found by health check-ups required by their work units. Those suspected patients were initially referred to the CPC for diagnosis, and the first X-ray and sputum smear test were free for them. This could reduce the delay of diagnosis and number of times visiting health-care facilities, as a result, less costs occurred during this phase. Inefficiencies in public health facilities and control services at private facilities are two main reasons for prolonged delay and increased costs of TB diagnosis [27, 28]. We found number of times visiting health-care facilities and hospitalization was one of the factors affecting costs due to TB diagnosis, with a mean of 3 visits to different health-care facilities. Only 35.6% of patients reported that they were diagnosed with suspected TB at their first visit, and the rest were informed that they had pneumonia, pleurisy, upper respiratory infection, and so on. Consistent with our study, other investigators pointed out that early diagnosis of TB reduced costs while achieving treatment success [29, 30]. The total number of registered active TB cases were 1317, included 76(5.8%) native case and 1241(94.2%) migrant cases in Bao'an district in 2013 [31]. In our study, 95% were migrant cases, and about 80% of patients were

factory workers or service staffs. We found that migrant patients faced much costs during TB care. This may be owing to the different modes of payment and free strategy. We find that patients who were better educated and occupied in national civil servant/services or retired staff may produce more costs before diagnosis. Those patients may have a good economic condition and care more about their health condition. The amount of sputum bacteria is an important factor affecting the outcome of treatment. Studies have shown that the higher the amount of sputum bacteria in the first sputum test, the worse effectiveness of treatment [32]. We find that sputum smear positive patients may have higher expenditures during TB treatment episodes due to a longer treatment period and more complicated conditions.

### 4.3 Policy and practice implications

Some measures can be taken to reduce the high financial burden of TB patients and to improve the performance of TB control programme. Given that under the current DOTS strategy in China, TB case were detected through passive case finding method, it is imperative to build an efficient coordination mechanism between referral hospitals and national TB programme. Further, it is also urgent to strengthen the consciousness and capacity of grass-roots public facilities to transfer suspected TB patients to better-equipped hospitals through improved work motivation to shorten the pathway for TB diagnosis and reduce the costs. Finally, active case finding may be shorten the delay of diagnosis, to reduce costs.

Migrant populations were less likely to be covered by social insurance, which might cause delays due to their health seeking behaviors [33] and more costs occurred during treatment, Hence, the migrant needs to be treated equally under social health coverage. Some subsidies provided to migrant tuberculosis may be effectively enhance the treatment compliance and reduce the costs owing to poor therapeutic effect [34].

Lacking of low ledge about tuberculosis symptoms and policy of TB prevention and treatment was the common factors related patients delay [35]. Those who visited more times to health facilities before diagnosis incurred more economic burden. It is necessary to strengthen publicity and education, especially among the migrant population.

### 4.4 Strengths and limitations

Fewer studies have measured the costs from the patients' perspective or the full costs, including direct and indirect costs, from the onset of symptoms to the end of treatment in south-east China. This study explored the full costs due to TB diagnosis and treatment. It is necessary to help decision makers understand the patient's economic situation and formulate a reasonable policy to control tuberculosis. In addition, the study population is representative of most TB patients in Shenzhen City, because about 50% of all cases under DOTS came from the study site. The re-treatment rate in our study was 5.84%, which is similar to that reported in other studies, with a re-treatment rate of 6.7% in Shenzhen [36] and 6.4% in other provinces in China [37]. This indicates that our study was acceptably representative of the treatment conditions for TB patients in Shenzhen specifically and China generally.

Our study does have some limitations. First, our study only investigated the economic burden brought by TB in a part of population. Those with severe co-morbidities such as AIDS, diabetes, and lung cancer and older than 59 were excluded. Therefore, our findings may be suitable for parts of TB patients who had the same condition as the cases in our study. Second, in our study, we did not survey the information regarding companion costs although it may not have much impact on the results since most TB patients in China did not need to be accompanied during the treatment. Third, we collected the range of the household's pre-TB annual income of every patient instead the exact number. As a result, we are not able to

conduct the analysis regarding catastrophic cost. Four, recall and reporting bias could not be avoided due to self-reported costs and personal income.

## 5. Conclusions

In conclusion, patients face heavy expenditures due to the costs of TB care, which accounted for 26% of patients' annual income. Costs were not equally distributed over time. In the TB treatment period, the costs are highest, exacerbating the problem of affordability. Free TB drugs are not the same as free treatment, and non-TB drugs, examinations, and loss of income are important considerations for patients. The income loss among all patients averaged 157 (57–855)USD. More visits to health-care facilities generated more out-of-pocket costs. Patients who had severe symptoms and re-treated patients bore a heavier burden. To avoid high costs, more attention should be paid to TB patients, even under the DOTS strategy. TB patients need testing methods with higher sensitivity, higher specificity, and lower costs to reduce their need for visits to health-care facilities. They also need more instructions to improve their adherence to treatment which may contribute to lower re-treatment rates.

## Supporting information

**S1 Table. Confounder adjusted association between costs due to TB diagnosis and various predictor variables using logistic regression models in the study of Bao'an district, Shenzhen City, China, 2013 (N = 514).**
(DOCX)

**S2 Table. Confounder adjusted association between *costs due to TB diagnosis* and various predictor variables using logistic regression models in the study of Bao'an district, Shenzhen City, China, 2013 (N = 514).**
(DOCX)

**S3 Table. Confounder adjusted association between *costs due to TB care* and various predictor variables using logistic regression models in the study of Bao'an district, Shenzhen City, China, 2013 (N = 514).**
(DOCX)

**S4 Table. Summary of independent predictors associated with costs due to TB diagnosis and treatment and care(diagnosis +treatment) in the study of Bao'an district, Shenzhen City, China, 2013 (n = 514).**
(DOCX)

**S1 Data.**
(XLS)

## Acknowledgments

The authors thank the participants who responded to our questionnaires and the TB doctors who participated in the questionnaire survey. We also express our deep gratitude to the staffs of the Bao'an Chronic Disease Prevention and Cure Hospital and the Center for TB Control of Guangdong Province.

## Author Contributions

**Conceptualization:** Yixiang Huang, Weiqing Chen.

**Data curation:** Liang Chen.

**Formal analysis:** Yixiang Huang, Jianying Huang, Xiaoting Su, Jianwei Guo, Weiqing Chen, Lingling Zhang.

**Funding acquisition:** Yixiang Huang, Weiqing Chen.

**Investigation:** Jianying Huang.

**Methodology:** Jianying Huang, Weiqing Chen.

**Project administration:** Yixiang Huang, Jianying Huang, Weiqing Chen.

**Resources:** Jianying Huang.

**Software:** Jianying Huang.

**Supervision:** Yixiang Huang, Weiqing Chen, Lingling Zhang.

**Validation:** Yixiang Huang, Jianying Huang.

**Visualization:** Jianying Huang, Xiaoting Su.

**Writing – original draft:** Yixiang Huang, Jianying Huang, Xiaoting Su.

**Writing – review & editing:** Yixiang Huang, Xiaoting Su, Liang Chen, Jianwei Guo, Lingling Zhang.

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
