## [Decision Letter · Decision Letter 0]

7 Apr 2020

PONE-D-20-06789

Analysis of the Economic Burden of Diagnosis and Treatment on Patients with Tuberculosis (TB) in the TB Control Demonstration Area of China

PLOS ONE

Dear Dr Yixiang Huang,

Thank you for submitting your manuscript to PLOS ONE. After careful consideration, we feel that it has merit but does not fully meet PLOS ONE’s publication criteria as it currently stands. Therefore, we invite you to submit a revised version of the manuscript that addresses the points raised during the review process.

Below you will find my comments along with comments of our two esteemed reviewers.

We would appreciate receiving your revised manuscript by 05 May 2020. To enhance the reproducibility of your results, we recommend that if applicable you deposit your laboratory protocols in protocols.io, where a protocol can be assigned its own identifier (DOI) such that it can be cited independently in the future. For instructions see: http://journals.plos.org/plosone/s/submission-guidelines#loc-laboratory-protocols

We look forward to receiving your revised manuscript.

Kind regards,

Hemant Deepak Shewade, MBBS MD

Academic Editor

PLOS ONE

Journal Requirements:

1. We note that you have indicated that data from this study are available upon request. PLOS only allows data to be available upon request if there are legal or ethical restrictions on sharing data publicly. For information on unacceptable data access restrictions, please see http://journals.plos.org/plosone/s/data-availability#loc-unacceptable-data-access-restrictions.

Additional Editor Comments (if provided):

Dear Authors,

In addition to the valuable points raised by the two reviewers, i have some comments from my end (some of these may be made by the reviewers as well)

1. Abstract - most were from the migrant population - why use ' most' here while elsewhere the number % have been given?

2. out of pockets expenditure? Please be careful with use of phrases (point made by the reviewer as well). Are you looking at total costs due to TB care or out of pocket expenditure. Overall in your paper, no reference was made to the WHO TB patient cost survey handbook 2016. I see that the study was among patients registered in 2013 and the WHO ptient cost survey booklet was not out then. Hence, it will be nice ot have a table in methods with clear operational definitions used.

3. 2.2 Study design title should be renamed as 2.2 Study design and population

4. Anonymized data HAS to be shared and this is ESSENTIAL

5. In the ethics statement, please mention the ethics number and the date

6. Line 175-176 (regarding primary outcome) - this line is not clear

7. Line 191 - shouldn't it be the household's pre-TB annual income?

8. Line 204-06. Similar issues were faced by us in another study. Another analysis may be to classify the costs as catastrophic or not and then do a log binomial regression. We can present both the results. See the following paper for example, as to how the analysis was done: https://www.ncbi.nlm.nih.gov/pubmed/30865730

9. Line 206 - Bivariate means two outcomes. All the analyses presented in this paper are univariate which can be further classified into single variable (curde or unadjusted) and multivariable (adjsuted) analysis.

10. What about catastrophic costs, what about inequity in distribution of catastrophic costs. See this paper for example. https://www.ncbi.nlm.nih.gov/pubmed/30173603

11. All the table and figure title have to be standalone with details about time place and person. Check for footnotes for abbreviations. All tables should be formatted again to make them presentable. For ex n and % in separate columns. n column shuold be right indented, (%) column should be left indented and vertical line between n and % should then be removed. CHeck other tables as well.

12. Why don't the authors look separately at factors associated with pre-dx costs and post-diagnosis costs.

13. If the authors have outcome data, why not compare unfav out and catastrophic costs, after adjusting for potential confounders.

Reviewers' comments:

Reviewer's Responses to Questions

**Comments to the Author**

1. Is the manuscript technically sound, and do the data support the conclusions?

Reviewer #1: Yes

Reviewer #2: Yes

2. Has the statistical analysis been performed appropriately and rigorously? 

Reviewer #1: Yes

Reviewer #2: Yes

3. Have the authors made all data underlying the findings in their manuscript fully available?

Reviewer #1: No

Reviewer #2: Yes

4. Is the manuscript presented in an intelligible fashion and written in standard English?

Reviewer #1: Yes

Reviewer #2: Yes

5. Review Comments to the Author

Reviewer #1: This work is on an important area namely the economic burden due to Tuberculosis related to the EndTB strategy goal of Zero Catastrophic cost for households due to Tuberculosis. The author have drafted the manuscript well and included several important points needed for a cost analysis study.

The study utilised a follow-up methodology to capture costs during the treatment phase. The authors have also studied some factors related to costs.

While the study justification rightly draws support from the EndTB Goal of Zero Catastrophic Cost, the authors have not included Catastrophic Cost estimation in the paper, despite having all the necessary data for the calculation of the same. It is highly recommended that the authors may include catastrophic cost and accordingly the manuscript may be redrafted. If there are data available on the coping costs or mechanism, they may also be included to understand the complete picture of the economic burden faced by the patients with Tuberculosis.

Abstract

Though the study includes both direct and indirect costs, the abstract starts with the terminology ‘expenditures’ which may apparently mislead a casual reader

Follow-up method utilised may be included in the methods under the abstract

Methodology

In the study settings,

the following may be included, which have implications while interpreting of the results: Proportion of Treatment and Retreatment Cases in the District; Actual figures of Incidence and Prevalence of Tuberculosis in Shenzhen Region/Bao’an District (lines 119-120), which needs to be specified seperately further between natives and migrants; No, of TB Cases in the District during the study year among natives and migrants; No. of (mandatory)visits as per the existing TB program during treatment; Proportion of patients with TB having co-morbidities namely diabetes, HIV, lung cancer etc.; Whether Active Case finding strategy is implemented or not; Mean/Median household (and/or Individual) income in the region/district; Proportion of people in different economic categories; a brief description of occupational health care provisions/model in the district relevant to TB care; brief description of the existing intervention in the district to address economic burden faced by patients/households due to TB; whether hospitalisation costs due to TB for persons with TB are covered under the existing program?;

Study Design:

Why were those aged above 59 years not included for the study? Reasons may be stated and also included/addressed in the discussion

In the discussion, need to consider, how non-inclusion of those with morbidity namely Diabetes, HIV affected the representativeness of the study. HIV and Diabetes have programmatic implications for Tuberculosis control as well as have been shown by studies in the literature to drive the costs due to tuberculosis higher.

Where those with MDR-TB excluded? Or is it that there were no drug resistant cases during the study period, which is quite unlikely. The authors have not stated whether the study is about costs due to drug sensitive tuberculosis. These needs to be clarified in title, abstract, justification (objectives), methods, and discussion.

Sample Size has not been estimated by the authors. Studies in the literature on Cost of illness have adopted different methods to calculate sample size. They may be referred to.

The cost data was based on self-reporting. The authors have mentioned both in the methodology and strengths in discussion, that that a logic-check was done. Brief description of this may be provided to help readers understand how this enabled better data quality

Timing of the first interview in the intensive phase could have affected the recall leading to differential recall among different participants. The data on delay between start of treatment and interview may be provided as mean and range. Similarly, it would be important to state whether the timing of multiple interviews were standardised for all patients.

Methodology can include explicitly the perspective from which cost was calculated, though it is understandable from the objective.

The reason for non-inclusion of guardian (companion) cost is not acceptable as the need for guardian are often higher during pre-diagnosis phase as the patients have more illness and may often make multiple visits to one or more provider before the diagnosis and start of treatment. Authors have also reported hospitalisations during pre-diagnosis period. Guardian (companion) costs needs to be included in the analysis.

The source of questionnaire may be stated and provided.

Though the study reported insurance coverage for participants, authors have not stated how reimbursements were adjusted against the costs in the analysis.

Results

Clinical characteristics including co-morbidities, pulmonary/extrapulmonary, etc to be included.

Care-seeking characteristics should include number of consultations/visits prior to diagnosis, number of providers visited and type of providers / facility visited (especially private providers /facility as it can vary from single doctor provider facility to corporate hospital)

Data on whether the participant was a beneficiary of any of the existing interventions in the district to reduce the economic burden to patients due to TB, needs to be reported, including the quantum of benefit

The cost during different phases of treatment could also be made available especially since authors have conducted multiple interviews through follow-up to obtain costs through the treatment period. They are important while planning interventions

Where there no hospitalisation episodes among participants during the treatment phase?

In Table 2, what are these Non-Tb drugs?

In table 3, what timeline does sputum smear test pertain to? What is the definition of household economic burden? What is the definition for degree of symptoms at diagnosis? Is it self-reported or based on clinical assessment? The Operational definition of these variables need to be included in the methodology.

Why insurance coverage was not included as a factor for economic burden?

Among the factors considered for economic burden, some of them pertain to pre-diagnosis costs and other may pertain to treatment phase costs and few many be for total costs. It is recommended that these be split accordingly instead of considering everything together, as there are implications for suggesting possible interventions to reduce cost.

Discussion

While the population of the district has 65% migrants, in the study, 95% of the study participants are migrants. This can have implications for representativeness of the study. This needs to be discussed in detail; this again importance further since it is one of the factors found to be significantly associated with economic burden as in table 3.

Socio-demographic and clinical characteristics of those 19 participants lost to follow-up need to be reported and the implications of the study results need to be discussed

Of the 533 participants, 176 patients were reported to have been shifted to CPC with no costs for patients during pre-diagnosis period. Can description of these transfers provided and included in study setting. How this lead to no delays and no costs are important for understanding the drivers of costs/economic burden.

Line 413, what are the author establishing by stating ‘recall and reporting bias could not be avoided, even in this longitudinal design’ The statement is self-contradictory. May need to be revised

Figure may not be needed as this information is represented in the table 2 and is clearly evident too.

Some Typographical Errors

Alignment of 70% in Age category 40-50 in Table 1

Subtitle Insurance to be made bold in Table 1

Weblink in line 112 may be shifted to reference and reference number provided

Title of Table 3, CNY needs to be removed

Reviewer #2: 1. The author needs to add some more details on opportunity cost i.e, income loss by patients due to illness. Some patients were hospitalized but they have earnings from job without wage loss then how to quantify the opportunity cost? Also, variation in salary is very heterogenetic. The author needs to describe about opportunity cost.

2. Under background characteristics, author mentioned about insurance company, but for Out of Pocket Expenditure (OOPE), we should not mention reimbursements given by insurance company. If the patients are getting reimbursements from any type of insurance, it should be deducted from the total out of pocket expenditure.

Net out of pocket expenditure = Total expenditure - Reimbursement.

Reimbursement amount should be highlighted if possible.

3. In methodology section, the author has mentioned the Chi-square test and Mann-Whitney analysis to examine the effect of each predictor variable on economic burden. But Chi-square test is used to know the association between predictor variables and regression analysis is used to understand the effects of predictor variables.

4. Table 3 analysis indicated that factors related to direct and opportunity costs (CNY) and in relation with the background variables. The heading should be association or differentials by background characteristics of patients and table 4 is showing the factor with regression analysis.

5. In table 4 multivariate analysis, only few predictor variables have been added. However, the author described in methodology section A forward stepwise approach was used to find the appropriate model. If P-value less than 0.05 is significant. It is good and recognised methods for adding the variables in regression analysis. Try also Variance inflation factor (VIF) to know the multicollinearity in predictor variables. If VIF less than 10, than the author can add the predictor variables and explore more variable as predictor variables.

Based on the above observations and looking into overall merit of the paper, few modifications in line of above points may be considered before sending it for publication.

6. PLOS authors have the option to publish the peer review history of their article (what does this mean?). If published, this will include your full peer review and any attached files.

Reviewer #1: No

Reviewer #2: Yes: Dr. Jeetendra Yadav

---

## [Author Response · Author response to Decision Letter 0]

18 May 2020

We have upload the file "Response to editor and reviewers" as attached file, please check for clearer and more detailed responses . The following is the copy of the file:

Additional Editor Comments (if provided):

Dear Authors,

In addition to the valuable points raised by the two reviewers, i have some comments from my end (some of these may be made by the reviewers as well)

1. Abstract - most were from the migrant population - why use ' most' here while elsewhere the number % have been given?

Thanks for the editor’s suggestion. We have replaced ‘most’ with the exact number ‘95%’ in the abstract, which was mentioned in the line 255.

2. out of pockets expenditure? Please be careful with use of phrases (point made by the reviewer as well). Are you looking at total costs due to TB care or out of pocket expenditure. Overall in your paper, no reference was made to the WHO TB patient cost survey handbook 2016. I see that the study was among patients registered in 2013 and the WHO ptient cost survey booklet was not out then. Hence, it will be nice ot have a table in methods with clear operational definitions used.

We are grateful for your suggestion. In our study, we did not include the cost covered by the medical insurance, hence it is inappropriate to use the term ‘out of pockets expenditure’. So we replaced it with ‘cost’ to make it more accurate. There are many kinds of medical insurance in China, and the patients may not be aware of whether they have certain kind of medical insurance or not, so they did not use the medical insurance to reduce their expenditure. However, in our study, the cost is measurable and comparable, so we revised it throughout this article accordingly.

According to your advice, we have added the operational definitions in table 1.

3. 2.2 Study design title should be renamed as 2.2 Study design and population

Revised, thank you.

4. Anonymized data HAS to be shared and this is ESSENTIAL

Thanks for this suggestion. We have provided the anonymous data as attachment in an .xls document.

5. In the ethics statement, please mention the ethics number and the date

We are grateful for the Reviewer’s suggestion. The ethical approval number is 2012-SPH021, date 9/28/ 2012.

6. Line 175-176 (regarding primary outcome) - this line is not clear

Thanks for this suggestion. We revised the definition to make it clear in line 192-193 as follows: The primary outcome variables were the mean (median) expense incurred during illness.

7. Line 191 - shouldn't it be the household's pre-TB annual income?

We appreciate your suggestion. Our study does have the limitation that we collected the range of the household's pre-TB annual income of every patient instead the exact number. However, we collected the exact number of the total annual income of every patient and as a result, we used the indicator to measure the patients’ economic burden.

8. Line 204-06. Similar issues were faced by us in another study. Another analysis may be to classify the costs as catastrophic or not and then do a log binomial regression. We can present both the results. See the following paper for example, as to how the analysis was done: https://www.ncbi.nlm.nih.gov/pubmed/30865730

We appreciate this suggestion. As mentioned above, we did not consider the catastrophic cost as an indicator in the design phase of our study, so we are not able to conduct the analysis regarding catastrophic cost. In our revision, based on the paper given above, we improved our method of statistics by adopting 2 methods in our revised manuscript.

9. Line 206 - Bivariate means two outcomes. All the analyses presented in this paper are univariate which can be further classified into single variable (curde or unadjusted) and multivariable (adjsuted) analysis.

Thanks for this suggestion. We have adopted a new method in the data analysis which is described carefully in line 221-244.

10. What about catastrophic costs, what about inequity in distribution of catastrophic costs. See this paper for example. https://www.ncbi.nlm.nih.gov/pubmed/30173603

We appreciate your suggestion. However, we did not adopt this professional term at the start of the study, and the annual patient income is expressed as a range instead of an exact number. As a result, it would be impossible to calculate the catastrophic costs to patients. 

11. All the table and figure title have to be standalone with details about time place and person. Check for footnotes for abbreviations. All tables should be formatted again to make them presentable. For ex n and % in separate columns. n column shuold be right indented, (%) column should be left indented and vertical line between n and % should then be removed. CHeck other tables as well.

We appreciate your suggestion. We have reformatted the tables throughout the article according to your suggestions.

12. Why don't the authors look separately at factors associated with pre-dx costs and post-diagnosis costs.

Thank you for this suggestion. We have analyzed the factors associated with different period of the treatment according to your suggestion and revised the statement in the data analysis, results and discussion section on the basis of the new analysis results.

13. If the authors have outcome data, why not compare unfav out and catastrophic costs, after adjusting for potential confounders.

We appreciate your suggestion. However, we did not adopt this professional term at the start of the study, and the annual patient income is expressed as a range instead of an exact number. As a result, it would be impossible to calculate the catastrophic costs to patients. 

Reviewer #1: This work is on an important area namely the economic burden due to Tuberculosis related to the EndTB strategy goal of Zero Catastrophic cost for households due to Tuberculosis. The author have drafted the manuscript well and included several important points needed for a cost analysis study.

The study utilised a follow-up methodology to capture costs during the treatment phase. The authors have also studied some factors related to costs.

While the study justification rightly draws support from the EndTB Goal of Zero Catastrophic Cost, the authors have not included Catastrophic Cost estimation in the paper, despite having all the necessary data for the calculation of the same. It is highly recommended that the authors may include catastrophic cost and accordingly the manuscript may be redrafted. If there are data available on the coping costs or mechanism, they may also be included to understand the complete picture of the economic burden faced by the patients with Tuberculosis.

We appreciate your suggestion. However, we did not adopt this professional term at the start of the study, and the annual patient income is expressed as a range instead of an exact number. As a result, it would be impossible to calculate the catastrophic costs to patients. 

In our study, most of the study population were factory workers. Only 186 workers knew that they had medical insurance and 305 workers knew that they did not have any medical insurance. Other 23 workers were unaware of whether they had medical insurance or not, and at the end of the study, they did not bring about any medical care reimbursement cost. Only 34 patients provided information about medical care reimbursement cost. As a result, the proportion of cost paid by medical insurance was not included in the analysis.

Abstract

Though the study includes both direct and indirect costs, the abstract starts with the terminology ‘expenditures’ which may apparently mislead a casual reader

Follow-up method utilised may be included in the methods under the abstract

Thank you for this suggestion. We have replaced the word ‘expenditures’ with ‘costs’ to avoid misunderstanding and added the follow-up method in the abstract.

Methodology 

In the study settings,

the following may be included, which have implications while interpreting of the results: 

1.Proportion of Treatment and Retreatment Cases in the District; 

Thanks for this suggestion. In the discussion section, we mentioned the proportion of treatment and retreatment cases in line 508-509 as follows: The re-treatment rate in our study was 5.84%, which is similar to that reported in other studies, with a re-treatment rate of 6.7% in Shenzhen and 6.4% in other provinces in China.

2.Actual figures of Incidence and Prevalence of Tuberculosis in Shenzhen Region/Bao’an District (lines 119-120), which needs to be specified seperately further between natives and migrants; 

We appreciate your suggestion. We have added the data in line 139-141 as follows: In 2010, the prevalence rate of smear positive of native population was 19.44 per 100 000 population, and that of the migrant population was 59.96 per 100 000 population.

3. No, of TB Cases in the District during the study year among natives and migrants; 

Thanks for this suggestion. We have listed the number in line 400-402 as follows: The total number of registered active tuberculosis cases were 1317, included 76(5.8) native case and 1241(94.2%) migrant cases in 2013.

4. No. of (mandatory)visits as per the existing TB program during treatment; 

Thank you for this suggestion. We have added the data in line 157.

5.Proportion of patients with TB having co-morbidities namely diabetes, HIV, lung cancer etc

We appreciate your suggestion. Because in our study, the patients with co-morbidities were excluded from the study, it is impossible to calculate the proportion of patients with TB having co-morbidities.

6.Whether Active Case finding strategy is implemented or not; 

We appreciate your suggestion. In the study setting section, we have added the statements in line 132-139 to clarify the way of TB patients being found in China as follows: The hospitals were the main place finding tuberculosis case and the Center for Prevention and Cure of chronic diseases (CPC) were the local designated TB facilities in the national TB programme networks, providing treatment and managing the tuberculosis cases. The healthcare providers in general hospitals were obligate to refer the suspected cases to CPC and the basic information of suspected cases were sent to CPC. If the cases did not consulted to CPC in time, the workers of CPC would contact the suspected case. Actively cooperating with medical institutions and TB prevention and control institutions, the discovery rate and report quality was increasing by years.

7.Mean/Median household (and/or Individual) income in the region/district; 

Thank you for this suggestion. We have added the median household income in Bao’an district in line 155-157 as follows: The mean household income in the region was 6,656.79 USD of 2012 according to the Statistical yearbook 2013. 

8. Proportion of people in different economic categories;

We appreciate your suggestion. However, we did not have access to the detailed information about the income of different population.

9. a brief description of occupational health care provisions/model in the district relevant to TB care; 

Thanks for this suggestion. In China, there is no occupational health care provisions/model focus on TB prevention only, the workers in our study were provided with the general occupational health service. For example, the workers may participant in the regular medical examination paid by the factory owner.

10. brief description of the existing intervention in the district to address economic burden faced by patients/households due to TB

We appreciate your suggestion. In line 148-151, we described the existing free policies to relief the economic burden on TB patients: The Center for Prevention and Cure of chronic diseases (CPC) is the institution authorized to provide TB diagnosis, treatment, and monitoring. The CPC provides patients with a radiologic imaging studies (X-ray) while diagnosed, and sputum smear tests during treatment, and anti-TB drugs for 6-8 months for free.

11. whether hospitalisation costs due to TB for persons with TB are covered under the existing program?

Thank you for this suggestion. The hospitalization costs due to TB was not covered under the existing program and born by the patients themselves. We have added the statements in line 148-151 to state the range of free TB policy. Since the cost regarding hospitalization is not included in the policy, it have to be borne by patients themselves.

Study Design:

Why were those aged above 59 years not included for the study? Reasons may be stated and also included/addressed in the discussion.

We appreciate your suggestion. People ages above 59 are more likely to have comorbidities which may influence the analysis of economic burden, so we excluded them. We have added the reason in line 166-167 as follows: The patients aged above 59 were excluded considering the elderly may have more commodities and those with MDR-TB were excluded too.

In the discussion, need to consider, how non-inclusion of those with morbidity namely Diabetes, HIV affected the representativeness of the study. HIV and Diabetes have programmatic implications for Tuberculosis control as well as have been shown by studies in the literature to drive the costs due to tuberculosis higher.

Thanks for this suggestion. We have discussed the influence of non-inclusion on the representativeness of our study in the limitation section in line 392-398 as follows: We included 533 eligible patients in the survey, 19 of which were lost to follow-up. The average age of them was 29.6±10.2years. The male was 14 and female was 5. One patient was with a severe symptoms appeared hemoptysis. The others were with mild or moderate symptoms like cough, chest distress and fever etc. the lost to follow-up was about 4 percent, and the general Socio-demographic and clinical characteristics of those 19 participants lost to follow-up were identical with the 514 patients. So，we considered there no much influence on our conclusion.

However, the exclusion can inevitably injure the representativeness, so we added this point as a limitation of our study in line 519-523 as follows: Our study only investigated the economic burden of patients brought by TB and those with severe co-morbidities such as AIDS, diabetes, and lung cancer were excluded. Therefore our study may reflect the economic burden faced by TB patients instead of all population.

Where those with MDR-TB excluded? Or is it that there were no drug resistant cases during the study period, which is quite unlikely. The authors have not stated whether the study is about costs due to drug sensitive tuberculosis. These needs to be clarified in title, abstract, justification (objectives), methods, and discussion.

Thank you for this suggestion. We have added the statement in the Study Design and population part in line165-167. To illustrate the study population of our survey more accurate, we have added the particular explanation in every section accordingly.

In our study, there were only 4 patients with MDR-TB among total 514 patients. One patient failed to achieve TB suppression in the third month of treatment, and the other three patients developed drug resistance during the treatment. Doctors adjusted the medicine according to their condition and part of second-line medicine is not free. Because there were few patients with MDR-TB, we did not eliminated them from the analysis. However, they may not have much impact on the stability of the analysis.

Sample Size has not been estimated by the authors. Studies in the literature on Cost of illness have adopted different methods to calculate sample size. They may be referred to.

We appreciate your suggestion. Ahead of our study, we did not calculate the sample size we needed. We included every patient who met the inclusion criteria and agreed to take part in our survey during the study period. Analysis results showed that the sample size of our study is enough to achieve the power required.

The cost data was based on self-reporting. The authors have mentioned both in the methodology and strengths in discussion, that that a logic-check was done. Brief description of this may be provided to help readers understand how this enabled better data quality.

We appreciate your suggestion. In the Data collection section, we have explained the process of a logic-check in line 187-189 as follows: Those contradictions or missing information would be asked of patients again in a subsequent interview. We also evaluated the quality of collected data by verifying the routine health information system.

Timing of the first interview in the intensive phase could have affected the recall leading to differential recall among different participants. The data on delay between start of treatment and interview may be provided as mean and range. Similarly, it would be important to state whether the timing of multiple interviews were standardised for all patients.

Thank you for this suggestion. First interview is in the first month of intensive phase treatment. We have added the median delay time of diagnosis in line 270-271. In the limitation section, we have mentioned that recall bias could not be avoided in our study as it relied on the information self-reported by patients. Patients went to see the doctor in chronic disease station nearby on time and were followed-up during their visit.

Methodology can include explicitly the perspective from which cost was calculated, though it is understandable from the objective.

Thanks for this suggestion. We have added the illustration in the line 192 as follows: We analysis the economic burden from tuberculosis patients’ perspective.

The reason for non-inclusion of guardian (companion) cost is not acceptable as the need for guardian are often higher during pre-diagnosis phase as the patients have more illness and may often make multiple visits to one or more provider before the diagnosis and start of treatment. Authors have also reported hospitalisations during pre-diagnosis period. Guardian (companion) costs needs to be included in the analysis.

We appreciate your suggestion. We regret mentioning that the companion costs was not surveyed in the study. In China, TB is a common disease and most patients do not need companions during the treatment. Another reason is that patients in our study were surveyed in the community or chronic disease station near their home within 15 minutes’ walk. So the companion costs during pre-diagnosis period may not have much impact on the outcome. We admit that it is a limitation of our study and have added the explanation in line 523-525 as follows: Second, in our study, we did not survey the information regarding companion costs although it may not have much impact on the results since most TB patients in China did not need to be companion during the treatment.

The source of questionnaire may be stated and provided.

Thanks for your suggestion. In this survey, we combined the aim of our study with standardized questionnaire used to investigate the demographic characteristic and designed the questionnaire. We have provided the questionnaire we used as attachment.

Though the study reported insurance coverage for participants, authors have not stated how reimbursements were adjusted against the costs in the analysis.

Thanks for your suggestion. In our study, most of the study population were factory workers. Only 186 workers knew that they had medical insurance and 305 workers knew that they did not have any medical insurance. Other 23 workers were unaware of whether they had medical insurance or not, and at the end of the study, they did not bring about any medical care reimbursement cost. Only 34 patients provided information about medical care reimbursement cost. As a result, the proportion of cost paid by medical insurance was not included in the analysis.

Results

Clinical characteristics including co-morbidities, pulmonary/extrapulmonary, etc to be included.

Care-seeking characteristics should include number of consultations/visits prior to diagnosis, number of providers visited and type of providers / facility visited (especially private providers /facility as it can vary from single doctor provider facility to corporate hospital)

We appreciate your suggestion. In the Socio-demographic Characteristics and Care-seeking Behaviors section, we have added the number of consultations prior diagnosis and type of facility visited in line 267-270. However, we did not collect information about the number of providers visited.

Data on whether the participant was a beneficiary of any of the existing interventions in the district to reduce the economic burden to patients due to TB, needs to be reported, including the quantum of benefit

Thanks for your suggestion. We have mentioned the coverage of the existing free TB policy in the Study setting section in line 149-151. In the second paragraph, we have also added the description of the extent of free treatment and the cost covered by the policy in line 404-406 as follows: Under the DOTS program, the CPC provide tuberculosis patients with a radiologic imaging studies (X-ray) , sputum smear tests and anti-TB drug during treatment for free amounting to 102.07 USD.

The cost during different phases of treatment could also be made available especially since authors have conducted multiple interviews through follow-up to obtain costs through the treatment period. They are important while planning interventions

Where there no hospitalisation episodes among participants during the treatment phase?

We appreciate your suggestion. Our study only investigated the cost reported by the patients occurred in the chronic disease station. There was no inpatient ward in the chronic disease station, so we did not surveyed the information about hospitalization.

In Table 2, what are these Non-Tb drugs?

Thank you for this suggestion. We have added the subsidiary drugs in line 152-153.

In table 3, what timeline does sputum smear test pertain to? What is the definition of household economic burden? What is the definition for degree of symptoms at diagnosis? Is it self-reported or based on clinical assessment? The Operational definition of these variables need to be included in the methodology.

Thanks for this suggestion. We have illustrated these definitions clearly in table 1.

Why insurance coverage was not included as a factor for economic burden?

Thanks for your suggestion. In our study, most of the study population were factory workers. Only 186 workers knew that they had medical insurance and 305 workers knew that they did not have any medical insurance. Other 23 workers were unaware of whether they had medical insurance or not, and at the end of the study, they did not bring about any medical care reimbursement cost. Only 34 patients provided information about medical care reimbursement cost. As a result, the proportion of cost paid by medical insurance was not included in the analysis.

Among the factors considered for economic burden, some of them pertain to pre-diagnosis costs and other may pertain to treatment phase costs and few many be for total costs. It is recommended that these be split accordingly instead of considering everything together, as there are implications for suggesting possible interventions to reduce cost.

We appreciate this suggestion. Considering everything together is truly not legible and easy to guide practice. So we decided to analyze the factors associated with pre-diagnosis and post-diagnosis separately according to your suggestion.

Discussion

While the population of the district has 65% migrants, in the study, 95% of the study participants are migrants. This can have implications for representativeness of the study. This needs to be discussed in detail; this again importance further since it is one of the factors found to be significantly associated with economic burden as in table 3.

Thanks for this suggestion. We have discussed it carefully in line 402-405 as follows: As one of the most population of Shenzhen City and owing numbers of factories , the migrant population in Bao’an district always exceeds the average level of Shenzhen City. The total number of registered active tuberculosis cases were 1317, included 76(5.8%) native case and 1241(94.2%) migrant cases in 2013. In our study, 95 percent were migrant cases, so our study can reflect the TB condition of Bao’an district truly.

Socio-demographic and clinical characteristics of those 19 participants lost to follow-up need to be reported and the implications of the study results need to be discussed

We appreciate this suggestion. In line 395-401, we have added the statement of the influence of lost follow-ups on our outcome. 

Of the 533 participants, 176 patients were reported to have been shifted to CPC with no costs for patients during pre-diagnosis period. Can description of these transfers provided and included in study setting. How this lead to no delays and no costs are important for understanding the drivers of costs/economic burden.

Thank you for this suggestion. We have added the detailed illustration in the study setting section in line 132-139.

Line 413, what are the author establishing by stating ‘recall and reporting bias could not be avoided, even in this longitudinal design’ The statement is self-contradictory. May need to be revised

Revised, thank you.

Figure may not be needed as this information is represented in the table 2 and is clearly evident too.

We appreciate this suggestion. We deleted figure 1 in the article.

Some Typographical Errors

Alignment of 70% in Age category 40-50 in Table 1

Subtitle Insurance to be made bold in Table 1

Weblink in line 112 may be shifted to reference and reference number provided

Title of Table 3, CNY needs to be removed

Revised, thank you.

Reviewer #2: 1. The author needs to add some more details on opportunity cost i.e, income loss by patients due to illness. Some patients were hospitalized but they have earnings from job without wage loss then how to quantify the opportunity cost? Also, variation in salary is very heterogenetic. The author needs to describe about opportunity cost.

We appreciate this suggestion. We have describe the opportunity cost in table 1 carefully. If TB patients still worked during the hospitalization, then there was no wage loss and the wage loss was 0. 

2. Under background characteristics, author mentioned about insurance company, but for Out of Pocket Expenditure (OOPE), we should not mention reimbursements given by insurance company. If the patients are getting reimbursements from any type of insurance, it should be deducted from the total out of pocket expenditure.

Net out of pocket expenditure = Total expenditure - Reimbursement.

Reimbursement amount should be highlighted if possible.

Thank you for this suggestion. We realized that it was inappropriate to use the term ’out of pocket expenditure’ to describe the economic burden for in our study, the cost covered by medical insurance is not included in the analysis. In our study, most of the study population were factory workers. Only 186 workers knew that they had medical insurance and 305 workers knew that they did not have any medical insurance. Other 23 workers were unaware of whether they had medical insurance or not, and at the end of the study, they did not bring about any medical care reimbursement cost. Only 34 patients provided information about medical care reimbursement cost. As a result, the proportion of cost paid by medical insurance was not included in the analysis.

3. In methodology section, the author has mentioned the Chi-square test and Mann-Whitney analysis to examine the effect of each predictor variable on economic burden. But Chi-square test is used to know the association between predictor variables and regression analysis is used to understand the effects of predictor variables.

We appreciate your suggestion. We have revised the analysis method and Chi-square test is not used this time. So we deleted it.

4. Table 3 analysis indicated that factors related to direct and opportunity costs (CNY) and in relation with the background variables. The heading should be association or differentials by background characteristics of patients and table 4 is showing the factor with regression analysis.

We have revised the table titles according to your advice and thanks for your advice.

5. In table 4 multivariate analysis, only few predictor variables have been added. However, the author described in methodology section A forward stepwise approach was used to find the appropriate model. If P-value less than 0.05 is significant. It is good and recognised methods for adding the variables in regression analysis. Try also Variance inflation factor (VIF) to know the multicollinearity in predictor variables. If VIF less than 10, than the author can add the predictor variables and explore more variable as predictor variables.

Thank you for this suggestion. We have adjusted the data analysis method according to your suggestion. 

Based on the above observations and looking into overall merit of the paper, few modifications in line of above points may be considered before sending it for publication.

---

## [Decision Letter · Decision Letter 1]

5 Jun 2020

PONE-D-20-06789R1

Analysis of the Economic Burden of Diagnosis and Treatment on Patients with non-drug resistance Tuberculosis (TB) in the TB Control Demonstration Area of China

PLOS ONE

Dear Dr. Wei-Qing Chen,

Thank you for submitting your manuscript to PLOS ONE. After careful consideration, we feel that it has merit but does not fully meet PLOS ONE’s publication criteria as it currently stands. Therefore, we invite you to submit a revised version of the manuscript that addresses the points raised during the review process.

We look forward to receiving your revised manuscript.

Kind regards,

Hemant Deepak Shewade, MBBS MD

Academic Editor

PLOS ONE

Additional Editor Comments (if provided):

Dear authors, Please find attached** (PFA)** the revised manuscript pdf along with my comments. The comments have been provided as sticky notes in the pdf. Carefully go through the comments one by one. I have suggested some changes in the analysis. Please also improve the grammar and other typo errors.

Reviewers' comments:

Reviewer's Responses to Questions

**Comments to the Author**

1. If the authors have adequately addressed your comments raised in a previous round of review and you feel that this manuscript is now acceptable for publication, you may indicate that here to bypass the “Comments to the Author” section, enter your conflict of interest statement in the “Confidential to Editor” section, and submit your "Accept" recommendation.

Reviewer #2: All comments have been addressed

2. Is the manuscript technically sound, and do the data support the conclusions?

Reviewer #2: Partly

3. Has the statistical analysis been performed appropriately and rigorously? 

Reviewer #2: Yes

4. Have the authors made all data underlying the findings in their manuscript fully available?

Reviewer #2: Yes

5. Is the manuscript presented in an intelligible fashion and written in standard English?

Reviewer #2: No

6. Review Comments to the Author

Reviewer #2: (No Response)

7. PLOS authors have the option to publish the peer review history of their article (what does this mean?). If published, this will include your full peer review and any attached files.

Reviewer #2: Yes: Dr Jeetendra Yadav

---

## [Author Response · Author response to Decision Letter 1]

19 Jun 2020

Title

1. May remove 'non-drug resistance' from the title. In the study population may explain that you included people with TB (not known to be drug-resistant)

Thank you for this suggestion. We have removed 'non-drug resistance' from the title and explain it clearly in the study population section.

2. remove (TB) form title 

Revised, thank you.

3. may remove 'non-demonstration area' from the title and mention the exact province or district in the title

Revised, thank you.

4,Analysis of the economic burden of diagnosis and treatment on patients with tuberculosis in xxxx, China

We appreciate this suggestion. We have revised the title according to your suggestion.

Abstract

1. Still in the abstract i do not get the folllowing information.What proportion of housholds incurred catastrophic costs due to 

- TB diagnosis (pre-diagnosis)

- TB treatment (post-diagnosis)

- TB care (pre plus post)

You have not mentioned the factors associated with catastrophic costs.

We appreciate this suggestion. Our study does have the limitation that we collected the range of the household's pre-TB annual income of every patient instead the exact number. We did not consider the catastrophic cost as an indicator in the design phase of our study, so we are not able to conduct the analysis regarding catastrophic cost. According to the reviewers’ suggestion, we adopted two new analysis methods to make our results more convincing.

2. Expand abbreviations at first use. Once an abbreviation is used, continue using it instead of the full form.Please check this throughout the paper

Revised, thank you.

3. Factors associated with

Revised, thank you.

4. Give exact percentage and frequency in parenthesis

Revised, thank you.

5. Please main consistency pre-diagnosis direct post-diagnosis medical and 

non-medical. BUT direct post-diagnosis? Shouldn't it be post-diagnosis direct costs?

post-diagnosis direct costs? I am suggesting the following structure to your results narrative in abstract in line with my suggestions for results narrative in main text

To avoid confusion, i suggest the following three important costs and presenting their median values here

costs due to TB diagnosis

costs due to TB treatment

costs due to TB care (diagnosis and treatment)

Then mention the % contribution of direct medical, direct non-medical, indirect costs to the above three costs. Then mention the key factors for high costs (due to diagnosis, due to treatment, due to care (diag+treat)). While mentioning the factors, instead of mentioning the variable, mention the exact sub category that was associated with high costs. 

Thanks for this suggestion. To make the description more accurate, we have revised the narrative regarding the cost occurred before and after diagnosis as follows:

We changed the ‘pre-diagnosis costs’ into ‘costs due to TB diagnosis’, ‘post-diagnosis costs’ into ‘costs due to TB treatment’ and ‘total costs’ into ‘costs due to TB care (diagnosis and treatment)’.

6. Does this mean post-diagnosis or pre+post-diagnosis?

We appreciate this suggestion. It means pre+post-diagnosis and in this revision we have changed it into ‘costs due to TB care (diagnosis and treatment)’.

7. Not clear

Education being associated with pre-diagnosis costs. I am not clear whether those who are educated have more pre-diagnosis costs or those with not educated have more pre-diagnosis costs. In other words the diagnosis of association is not clear. 

Please may modify as follows.

Factors associated with more pre-diagnosis costs are: xxx, yyy, zzz. (just do not mention education, mention whether it is less education or more education)

Similarly mention: Factors association with more post-diagnosis costs are: xxx, yyy, zzzz Like education, clarify for household registration (ye or no), sputum smear status (high grade or low grade, marital status (married or unmarried)

We appreciate this suggestion. We have modified the illustration as follows (line 55-60):

Those who visited more times to health facilities, hospitalized, received higher education or occupied in national civil servant/services /retired staff might expense more before diagnosis. Factors Costs due to TB treatment among migrant patients, Sputum smear positive patients, widowed/divorced population, was significantly higher. Factors associated with less total costs were native patients, fewer times of visiting to health-care facilities and those with no hospitalization history due to TB.

Background

1.What is the word count? from introduction to conclusion (excluding the tables and figures?). 

I see this article to be very long. Though plos one does not have a word count limit, the article should be made more crisp and limited to 3000 words (max)

Thanks for this suggestion. We revised the title ‘Introduction’ as ‘Background’. In order to state our results simpler and clear, we have deleted a lot of statement which is not pretty related to our research and put some tables as the supplement materials.

2.These have to be updated as per WHO Global TB report 2019 (data for year 2018)

Thanks for your suggestion and we have updated the data according to the Global TB report 2019.

3.Not clear

Thanks for this suggestion. To avoid confusion, we have deleted these words.

Materials and Methods

1.What about assessment of catastrophic costs and factors associated with catastrophic costs?

We appreciate your suggestion. As mentioned above, we did not consider the catastrophic cost as an indicator in the design phase of our study, so we are not able to conduct the analysis regarding catastrophic cost. We evaluated the economic burden as the proportion of patients’ total annual income instead.

2.Expand abbreviations on first use.

Revised, thank you.

3.How is this related to the current study?

We appreciate your suggestion. This part is not directly associated with our study and we have removed it from the article.

4.Is this relevant to the current study?

Removed, thank you.

5.Use 'previously treated' throughout the manuscript - narrative text, tables and figures

Revised, thank you.

6.TB patients (not known to be drug resistant)

Revised, thank you.

7.Transferred-in patients

Revised, thank you.

8.Mention this line in measurements and definition section. May be at the beginning of second para in measurements and definiton section. Remove it from here.

Thanks for your suggestion and we have removed it from here according to your advice.

9.What did you do when there was unfavourable outcomes, say loss to follow up? did you use monthly average of previous months and added it to the reminder of the treatment. What was done when a patient was transferred out? Transfer out is common among migrants.

We appreciate your suggestion. The present study surveyed 533 eligible patients, 19 of whom were lost to follow-up, one of whom died from another disease, and 18 of whom moved to other provinces, the lost rate nearly 4%. Considering the lost rate was low, those patients were eliminated directly in the analysis.

10.May start this para by refering to Table 1. May put the details of costs calculation in Table 1. May mention the salient features here to avoid duplication. What about calculation of catastrophic costs. It has not been included in narrative or in the table.

The details of costs calculation were showed in the table 1.

Revised, thank you.

11.May remove this.

Revised, thank you.

12.May mention (indirect costs) in brackets.

Revised, thank you.

13.What is TP?

Thanks for your suggestion. We made a typo and have revised it as ‘TB’.

14.I guess this may be removed as well.

Revised, thank you.

15.I think you used median, may state so.

Revised, thank you.

16.Statement not complete. Please check.

Revised, thank you.

17.Please mention the approval number / letter number and date in parenthesis.

We are grateful for your suggestion. We have added the ethical approval number in the article. 

Results

1.As mentioned before, the article is too long. One example is results narrative text. 

The results narrative text should focus on the key results which should be in line with the aim and objectives. For details, the authors may always refer to the tables. 

I suggest the authors to go through this paper for presentation of results. https://pubmed.ncbi.nlm.nih.gov/30173603/

Table 2 - baseline characteristics table 

(see table 2 of above paper)

Table 3 - Table summarising the break of costs, three columns, costs due to diagnosis, treatment, diagnosis+treatment (see table 3 of above paper)

Table 4 - Contribution of direct medical, direct non-medical and indirect costs to the costs (in %), three columns as in Table 3 ((see table 3 of above paper)

Table 5,6,7: Factors assoc with all three costs - total costs due to diagnosis,treatment, diagnosis and treatment. (one table for each). See this paper to understand how these tables should be formatted (see table 3-8) https://bmchealthservres.biomedcentral.com/articles/10.1186/s12913-018-3583-y

Then in Table 8, my summarise the factors. See table S1 (Additional file 1) in this paper.

https://bmchealthservres.biomedcentral.com/articles/10.1186/s12913-018-3583-y

Table five, six, seven may also be put as supplementary table as one cannot interpret the B coefficient much as the data is log transformed. I do not see the factors associated with total costs more than the median value (logistic regression). ?

This was one of my suggestions in previous round. In the revised version, the authors have mention this in data analysis section but have not incuded the same in results narrative. May compare the risk factors in linear regression and logistic regresion. Your results narrative may tell variables that are significanlty associated in both the models (and esp across diagnosis, treatment and diag+treat costs). Focus your discussion section on these variables. 

Thanks for your suggestion. We have read the reference carefully and rearranged all the tables in this article. We summarize the factors in the supplement material in Table S4.

We also added the analysis regarding the factors associated with total cost according to your suggestion.

2.While writing the results, main the following consistency either % (number) or number (%) or only %. Please do not mix, it becomes difficult to read.

Thank you for this suggestion. We have revised the description of results with %(number) throughout the article.

3.As shared before it is not clear what the final sample size of the study population 

is? The same sample size should be the denominator for all your analysis.

Previous table sample size was around, 500, here it is 338. This is not the correct way to present. If people did not incur direct costs, mention their costs as zero and include them in the analysis.

We appreciate your suggestion. In the new tables, we have included all the patients into the analysis.

4.Those who did not incur pre-diagnosis costs should also be included. In other words, the denominator for calculation of costs has to be the same.

Revised, thank you.

5.Denominator has to be consistent as 514. If someone did not have costs under a 

heading then consider it as zero and include it in the calculation of mean/median. See table 3 in https://pubmed.ncbi.nlm.nih.gov/30173603/

Instead of these details i would like to know in the form of a table or a 100% component bar diagram, what was the percentage contribution of direct medical, direct non-medical and indirect costs to total costs in the pre-diagnosis period, post diagnosis period and pre-post combined. See table 4 in https://pubmed.ncbi.nlm.nih.gov/30173603/

Thanks for your suggestion. We have added the table 4 to present the percentage contribution of direct medical, direct non-medical and indirect costs to total costs in the pre-diagnosis period, post diagnosis period and pre-post combined.

6.Not clear.when in hospital (ever? anytime during diagnosis?)

Thanks for your suggestion. We have revised the category as ‘Whether in hospital due to TB diagnosis’.

7.Only significant variables presented? if yes, clarify in footnotes. For B coefficient, mention in footnote that linear regression was done after conversion to log scale. Hence, cannot be directly interpreted. but significance can be assessed.

We are grateful for your suggestion. We present the results of logistics 

regression in the supplement table 1-3 and clarified it in the footnotes.

8.Predictor?

Revised, thank you.

9.Can exclude from table.

Revised, thank you.

10.E should be capital

Revised, thank you.

11.Similar comments as in Table 4. This table is well formated though

Revised, thank you.

---

## [Decision Letter · Decision Letter 2]

16 Jul 2020

PONE-D-20-06789R2

Analysis of the Economic Burden of Diagnosis and Treatment on Patients with Tuberculosis in Bao'an district of Shenzhen city, China

PLOS ONE

Dear Dr. Wei-Qing Chen,

Thank you for submitting your manuscript to PLOS ONE. After careful consideration, we feel that it has merit but does not fully meet PLOS ONE’s publication criteria as it currently stands. Therefore, we invite you to submit a revised version of the manuscript that addresses the points raised during the review process.

We look forward to receiving your revised manuscript.

Kind regards,

Hemant Deepak Shewade, MBBS MD

Academic Editor

PLOS ONE

Additional Editor Comments (if provided):

Dear Authors, Below are editor and reviewer comments

EDITOR COMMENTS

Table 2. Mention N=524 in title

Table 3 and 4 - format the column titles

Do not repeat the results data in discussion. Results section is "what we found". Discussion section is "what it means"

Discussion section - please have a relook considering the principles below

First para - do not repeat the results. First para should include summary of results narrative using simple language (Without numbers) in 3-4 lines. When in doubt about the key findngs, revert back to the objectives

Then each finding should be discussed in one para. If three key findings, then three para.

Then in next para, what are the policy/practicice implications and future research

Strenght / Limitation

Please feel free to have these subheadings in discussion section: Summary of key findings, Discussion of key findings, Policy and practice implications, Strengths and limitations

REVIEWER 3 COMMENTS (they are in the form of an attachement, i am copying them here as well)

OVERALL COMMENTS

The study provide useful information about the economic burden of TB cases in Shenzhen city in China, which reflect the key indicator of the End- TB strategy. In spit the data was not so fresh ,the method and the definition was not in line with the latest TB patient cost handbook issued by WHO, the results still have some reference value for the evaluate of the progress in the TB control in south of China.

SPECIFIC COMMENT

1． Line 41：to replace the sentence“Illness-related costs experienced by patients” with “Illness-related costs experienced by tuberculosis patients” may be more related to the title

2． Line 43: “ full costs, including direct and opportunity costs” and Line 87 “opportunity costs ”is not clear, we recommend to use indirect cost directly and you use it in Table1.

3. Line 96-97 the reference of the migrant population proportion and income was inconsistent, considering the study was carried out in 2013, the statistical yearbook 2013 will be better.

4. Line 104 “The hospitals”is not accurate, here the hospital means general hospital?

5. Line 114-115 the discription was not accurate, both the X-ray and sputum smear was carried out during diagnosis and treatment, please redescribe.

6. Line 119-120 The description of “The drugs prescribed every month including auxiliary examinations and subsidiary drugs like……. etc. are not free.” was obscure, may misleading reader that all the drugs including TB drugs was not free. Please clarify.

8. Line 125 “1 January 2013 to June 2013” should be modified to “1 January 2013 to 30 June 2013”

9. Line128: the reason your study excluded the cases older than 59 was considering they might have more comorbidities, yet in the exclude criterions already mentioned the severe comobidities, so if it was necessary to exclude the elderly? Please reconsider.

10. Line 161-163 the analysis did not included the income lost of companions for the few proportion and did not included the loss time for daily drug intake for the reason of most patients done this on their way to wort….., it would be better to clearly list the exact number replace “a few” or “most”, which will provide more exactly information to the reader. In my opinion , it was more rational to include this part in the analysis.

11. Line 199 : “19 of whom were lost to follow-up” and Line 201 “The rate of loss to follow-up was nearly 4% (19)” was repeat information, Please simplify after integration

12. Line 204-205 :“They were diagnosed at CPC immediately without consulting to other facilities, so their costs due to TB diagnosis were not calculated”. This information should be described in the method part instead of the results. Another question, for these cases the diagnosis fee should not be ignored considering they also should be confirmed diagnosis in the CPC.

13. Line 209: Under the title of Table 2, should give the N

14. Table 2 : Please recheck of the number and percentage of Age( years), the total number is only 401, please describe the age of other 103 cases, if the information was missing ,please clarify also.

15. Table 2: “self-reported economic status*” what was the meaning of *, please describe in the footnote.

16. Line 211: TB care( diagnosis +treatment), as you have defined the TB care in the method part, so it was not necessary to take the bracket here

17. Line 224 : Please use “indirect costs” to replace “opportunity costs” that will ensure the consistent with the describe in the Table3

18. The first line of Table3 and Table 4, Please use “TB care” to replace “TB diagnosis+treatment” and “TB care( diagnosis+treatment) ”

19. Please recheck the ß value in Line 241 Line 245

20. Line 246 Table6??

21. For the discussion part, there were some repeat information of the results, such as Line 290 …. please go through this part and focus on the “real discussion”

Reviewers' comments:

Reviewer's Responses to Questions

**Comments to the Author**

1. If the authors have adequately addressed your comments raised in a previous round of review and you feel that this manuscript is now acceptable for publication, you may indicate that here to bypass the “Comments to the Author” section, enter your conflict of interest statement in the “Confidential to Editor” section, and submit your "Accept" recommendation.

Reviewer #3: (No Response)

2. Is the manuscript technically sound, and do the data support the conclusions?

Reviewer #3: (No Response)

3. Has the statistical analysis been performed appropriately and rigorously? 

Reviewer #3: (No Response)

4. Have the authors made all data underlying the findings in their manuscript fully available?

Reviewer #3: (No Response)

5. Is the manuscript presented in an intelligible fashion and written in standard English?

Reviewer #3: (No Response)

6. Review Comments to the Author

Reviewer #3: (No Response)

7. PLOS authors have the option to publish the peer review history of their article (what does this mean?). If published, this will include your full peer review and any attached files.

Reviewer #3: No

---

## [Author Response · Author response to Decision Letter 2]

25 Jul 2020

For clearer version of Response to Reviewers, please check the uploaded file "Response to Reviewers 0725", thank you. The following is the copy of the uploaded file:

EDITOR COMMENTS

1.Table 2. Mention N=524 in title

Thanks for your suggestion and we have added the number in the title of table 2.

2.Table 3 and 4 - format the column titles

We are grateful for this suggestion. In the last round of revision, the editor provided us with an article to refer to and we have rearranged the format of table 3 and 4 as the tables in the article (The table in this picture is one example). We have also revised the first column title of these two tables according to your suggestion.

3.Do not repeat the results data in discussion. Results section is "what we found". Discussion section is "what it means"

Discussion section - please have a relook considering the principles below

First para - do not repeat the results. First para should include summary of results narrative using simple language (Without numbers) in 3-4 lines. When in doubt about the key findngs, revert back to the objectives

Then each finding should be discussed in one para. If three key findings, then three para.

Then in next para, what are the policy/practice implications and future research。

Strenght / Limitation

Please feel free to have these subheadings in discussion section: Summary of key findings, Discussion of key findings, Policy and practice implications, Strengths and limitations.

We are grateful for your suggestion and have reorganized the structure and content of the discussion section. We deleted the redundant description of results and added more summaries of the key findings.

REVIEWER 3 COMMENTS (they are in the form of an attachement, i am copying them here as well)

OVERALL COMMENTS

The study provide useful information about the economic burden of TB cases in Shenzhen city in China, which reflect the key indicator of the End- TB strategy. In spit the data was not so fresh ,the method and the definition was not in line with the latest TB patient cost handbook issued by WHO, the results still have some reference value for the evaluate of the progress in the TB control in south of China.

SPECIFIC COMMENT

1． Line 41：to replace the sentence“Illness-related costs experienced by patients” with “Illness-related costs experienced by tuberculosis patients” may be more related to the title

Thanks for this suggestion. We have revised the sentence according to your suggestion.

2． Line 43: “ full costs, including direct and opportunity costs” and Line 87 “opportunity costs ”is not clear, we recommend to use indirect cost directly and you use it in Table1.

Thank you for this suggestion and we have replaced the ‘opportunity costs’ with ‘indirect costs’ throughout this article.

3. Line 96-97 the reference of the migrant population proportion and income was inconsistent, considering the study was carried out in 2013, the statistical yearbook 2013 will be better.

We are grateful for this suggestion. We have revised the year and added the data of 2013 in the article.

4. Line 104 “The hospitals”is not accurate, here the hospital means general hospital?

Thank you for this suggestion. Under the current DOTS strategy in China, TB case were detected through passive case finding method. Most suspected cases were discovered during their treatment in general hospitals and then sent to Center for Prevention and Cure of chronic diseases (CPC) by the health-care providers. So, the general hospitals were the main place to find TB cases. In our article, we have revised it as general hospitals.

5. Line 114-115 the discription was not accurate, both the X-ray and sputum smear was carried out during diagnosis and treatment, please redescribe.

We are grateful for this suggestion. To make the description more accurate, we have revised the statement as follows(line 113-116): The CPC is the institution authorized to provide TB diagnosis, treatment, and monitoring, with a radiologic imaging studies (X-ray) and sputum smear tests while diagnosed for free ,and 5 sputum smear tests and anti-TB drugs for 6-8 months during treatment for free. 

6. Line 119-120 The description of “The drugs prescribed every month including auxiliary examinations and subsidiary drugs like……. etc. are not free.” was obscure, may misleading reader that all the drugs including TB drugs was not free. Please clarify.

We are grateful for your suggestion. We have revised this section and listed the drugs and service which are not free during the treatment clearly.

8. Line 125 “1 January 2013 to June 2013” should be modified to “1 January 2013 to 30 June 2013”

Revised. Thank you.

9. Line128: the reason your study excluded the cases older than 59 was considering they might have more comorbidities, yet in the exclude criterions already mentioned the severe comobidities, so if it was necessary to exclude the elderly? Please reconsider.

We are grateful for your suggestion. Besides the reason mentioned above, we also considered that the patients aged above 59 may have more have more respiratory system diseases and higher compliance which may affect the result of our research. To avoid this confounding factor, we excluded this age group. We have added this reason in the article.

10. Line 161-163 the analysis did not included the income lost of companions for the few proportion and did not included the loss time for daily drug intake for the reason of most patients done this on their way to wort….., it would be better to clearly list the exact number replace “a few” or “most”, which will provide more exactly information to the reader. In my opinion , it was more rational to include this part in the analysis.

We have used the exact number in this sentence and thank you for this suggestion.

11. Line 199 : “19 of whom were lost to follow-up” and Line 201 “The rate of loss to follow-up was nearly 4% (19)” was repeat information, Please simplify after integration

Thank you for this suggestion. We have revised the description as follows (line 205-207): The present study surveyed 533 eligible TB patients with no co-morbidity, 19 (4%) of whom were lost to follow-up, one of whom died from another disease, and 18 of whom moved to other provinces.

12. Line 204-205 :“They were diagnosed at CPC immediately without consulting to other facilities, so their costs due to TB diagnosis were not calculated”. This information should be described in the method part instead of the results. Another question, for these cases the diagnosis fee should not be ignored considering they also should be confirmed diagnosis in the CPC.

In our study, 176 patients were discovered during health check-ups required by their work units and the suspected cases were sent to the CPC. The CPC provided them with a radiologic imaging studies (X-ray) and sputum smear tests while diagnosed for free. As a result, these patients may not generate costs before diagnosis and we did not survey the costs occurred while diagnosed of this part of patients. 

13. Line 209: Under the title of Table 2, should give the N

Revised. Thank you.

14. Table 2 : Please recheck of the number and percentage of Age( years), the total number is only 401, please describe the age of other 103 cases, if the information was missing ,please clarify also.

Thank you for this suggestion. We feel sorry that we made a mistake in this part and we have corrected it with the right number.

15. Table 2: “self-reported economic status*” what was the meaning of *, please describe in the footnote.

We are grateful for your suggestion. Under the table 2, we have added the footnote to describe the meaning of the symbol.

16. Line 211: TB care( diagnosis +treatment), as you have defined the TB care in the method part, so it was not necessary to take the bracket here

Revised. Thank you.

17. Line 224 : Please use “indirect costs” to replace “opportunity costs” that will ensure the consistent with the describe in the Table3

Thank you for this suggestion and we have replaced the ‘opportunity costs’ with ‘indirect costs’ throughout this article.

18. The first line of Table3 and Table 4, Please use “TB care” to replace “TB diagnosis+treatment” and “TB care( diagnosis+treatment) ”

We are grateful for your suggestion. We have revised the first line of these two tables according to your suggestion.

19. Please recheck the ß value in Line 241 Line 245

Revised. Thank you.

20. Line 246 Table6??

Thank you for this suggestion. The results here is presented in table S1, so we deleted the table 6 here and stated the correct table S1 at the end of this paragraph. 

21. For the discussion part, there were some repeat information of the results, such as Line 290 …. please go through this part and focus on the “real discussion”

We are grateful for your suggestion. We have reorganized the structure and content of the discussion section according to your suggestion.

---

## [Editor Report · Decision Letter 3]

5 Aug 2020

Analysis of the Economic Burden of Diagnosis and Treatment on Patients with Tuberculosis in Bao'an district of Shenzhen city, China

PONE-D-20-06789R3

Dear Dr. Wei-Qing Chen,

We’re pleased to inform you that your manuscript has been judged scientifically suitable for publication and will be formally accepted for publication once it meets all outstanding technical requirements.

Kind regards,

Hemant Deepak Shewade, MBBS MD

Academic Editor

PLOS ONE
---

## [Editor Report · Acceptance letter]

17 Aug 2020

PONE-D-20-06789R3 

Analysis of the Economic Burden of Diagnosis and Treatment on Patients with Tuberculosis in Bao'an district of Shenzhen city, China 

Dear Dr. Chen:

I'm pleased to inform you that your manuscript has been deemed suitable for publication in PLOS ONE. Congratulations! Your manuscript is now with our production department. 

Kind regards, 

on behalf of

Dr. Hemant Deepak Shewade 

Academic Editor

PLOS ONE